# Gut Microbiota Reshaped by Pectin Treatment Improves Liver Steatosis in Obese Mice

**DOI:** 10.3390/nu13113725

**Published:** 2021-10-22

**Authors:** Camille Houron, Dragos Ciocan, Nicolas Trainel, Françoise Mercier-Nomé, Cindy Hugot, Madeleine Spatz, Gabriel Perlemuter, Anne-Marie Cassard

**Affiliations:** 1Université Paris-Saclay, Inserm U996, Inflammation, Microbiome and Immunosurveillance, 32 rue des carnets, 92140 Clamart, France; camille.houron@gmail.com (C.H.); ciocanelro@yahoo.com (D.C.); nicolas.trainel@universite-paris-saclay.fr (N.T.); Cindy.hugo60@gmail.com (C.H.); madeleine.spatz@hotmail.fr (M.S.); gabriel.perlemuter@aphp.fr (G.P.); 2AP-HP, Hepato-Gastroenterology and Nutrition, Hôpital Antoine-Béclère, 92140 Clamart, France; 3Université Paris-Saclay, Inserm, CNRS, Institut Paris Saclay d’Innovation Thérapeutique, 5 rue J.B. Clément, 92296 Châtenay-Malabry, France; francoise.mercier-nome@universite-paris-saclay.fr

**Keywords:** NAFLD, NASH, fecal microbiota transplantation, SCFA, fiber, pectin

## Abstract

Pectin, a soluble fiber, improves non-alcoholic fatty-liver disease (NAFLD), but its mechanisms are unclear. We aimed to investigate the role of pectin-induced changes in intestinal microbiota (IM) in NAFLD. We recovered the IM from mice fed a high-fat diet, treated or not with pectin, to perform a fecal microbiota transfer (FMT). Mice fed a high-fat diet, which induces NAFLD, were treated with pectin or received a fecal microbiota transfer (FMT) from mice treated with pectin before (preventive FMT) or after (curative FMT) being fed a high-fat diet. Pectin prevented the development of NAFLD, induced browning of adipose tissue, and modified the IM without increasing the abundance of proteobacteria. Preventive FMT also induced browning of white adipose tissue but did not improve liver steatosis, in contrast to curative FMT, which induced an improvement in steatosis. This was associated with an increase in the concentration of short-chain fatty acids (SCFAs), in contrast to preventive FMT, which induced an increase in the concentration of branched SCFAs. Overall, we show that the effect of pectin may be partially mediated by gut bacteria.

## 1. Introduction

Being overweight or obese is linked to an increased risk of metabolic syndrome, including insulin resistance, type 2 diabetes mellitus, mixed dyslipidemia, and non-alcoholic fatty-liver disease (NAFLD) [1]. Among overweight/obese patients, only a subset of individuals develop severe liver lesions, such as non-alcoholic steatohepatitis (NASH). The intestinal microbiota (IM) has been demonstrated to play a major role in the individual susceptibility of patients to developing NAFLD [2,3]. The large overlap between NAFLD and metabolic disorders may complicate the establishment of a specific bacterial signature associated with the disease spectrum. Nevertheless, changes in the composition of the IM are involved in NAFLD and/or the development of NASH, including an increase in the abundance of *Akkermansia*, *Dorea*, *Escherichia, Parabacteroides, Porphyromonas, Ruminococcus, Shigella,* and *Propionibacterium acnes*, and a decrease in that of *Alistipes, Faecalibacterium, Haemophilus* and *Clostridium coccoides*, and *Bacteroides fragilis* [4].

The composition of the IM in adults depends on several features, including primary colonization, antibiotic treatment, infections, environment, genetic polymorphisms, and, obviously, nutrients [5,6,7]. The western-style diet, rich in fat and sugar and low in fiber, contributes, at least partially, to the depletion of specific bacterial taxa in the IM [8,9]. Such alterations may contribute to an increase in the development of chronic inflammatory diseases, including those associated with obesity [10]. The western diet induces prolonged metabolic stress, leading to adipose tissue dysfunction, inflammation, and adipokine release, which trigger liver injury. The IM is partially involved in the metabolic disorders resulting from this diet [11]. Moreover, targeting the IM can both improve metabolic disorders and decrease adipose tissue inflammation by favoring the browning of white adipose tissue (WAT) [12].

Thus, the analysis of the bacterial signature in NAFLD is important for patient follow-up, and the restoration of a “healthy” IM could be relevant. Indeed, the impact of dietary fiber on gut microbiota has been clearly established [13]. It has been shown that fiber prevents liver injury, depending on the type; galactomannan worsens liver injury, whereas inulin or pectin may improve it [14,15,16]. Moreover, we have already demonstrated that pectin alters IM composition and improves alcohol-induced liver injury [17] by increasing the abundance of *Bacteroides*, bacteria for which the abundance is decreased in NAFLD. Nevertheless, fiber consumption can have side-effects, including abdominal discomfort and intestinal pain, which compromises its use in certain individuals. Understanding which bacteria are beneficial, specifically those involved in the protective effect of fiber, is important for the development of new probiotic treatments.

Here, we investigated the role of pectin and the IM in preventive and curative treatment of NAFLD. Fecal microbiota transfer (FMT) from mice treated with pectin showed that curative FMT induced an improvement in NAFLD associated with changes in the IM and the microbial metabolite short-chain fatty acids (SCFAs).

## 2. Materials and Methods

### 2.1. Mice and Diets

Five-week-old male C57BL/6 J mice were purchased from Janvier laboratory (Le Genest, France) and maintained under a 12 h light/dark schedule, with food and water ad libitum, and treated in accordance with the Guide for the Care and Use of Laboratory Animals (National Research Council, 1996). The animal experimentation procedure was validated by the local ethics committee, CEEA26 (APAFiS #02734.03, APAFiS #4255-2016080812089425v1 and APAFiS #20833). Mice were fed either a normal diet (ND) or a high-fat diet (HFD), with or without 2% pectin (0.06 g pectin/30 g of mouse), in which the energy content of fat was either 4.1% or 34.7%, respectively. The caloric content was 360.9 kcal/100 g for the ND, 351.3 kcal/100 g for the ND with 2% pectin (ND^2^), 516.2 kcal/100 g for the HFD, and 506.7 kcal/100 g for the HFD with 2% pectin (HFD^2^). The HFD diets were supplemented with lard (31.5%). To assess the role of pectin as a curative treatment, mice were fed an HFD for 16 weeks and received pectin supplementation in the HFD (2%) from week 16 to week 24 as a curative treatment (HFD + HFD2) versus mice who did not received pectin (HFD + HFD). Pectin, extracted from apples, was purchased from Sigma-Aldrich (P93854, lot: BCBS3576). All diets were manufactured by SNIFF (Germany). Mice were weighed every week and dietary intake was measured and averaged for each cage (four mice per cage).

### 2.2. Fecal Microbiota Transfer

Microbiota transfer was performed by feces gavage using a modified version of a previously described protocol [18]. Feces was recovered from 34 mice (C57BL/6 J fed either HFD, or HFD^2^), diluted in BHI (Brain Heart Infusion, Becton Dickinson) supplemented with 0.5 mg/mL L-cysteine (Sigma-Aldrich, St-Louis, MO, USA) and 20% skim milk (Becton Dickinson) (vol/vol), and stored in aliquots at −80 °C. Then, a 100 µL volume, containing 3.33 mg feces, was administered to each corresponding mouse twice a week for two months.

### 2.3. Glucose Tolerance

Oral glucose tolerance tests (OGTTs) were performed as follows: a glucose load (2 g/kg) was given by gavage after 6 h of fasting and blood samples were taken from the tail vein at 0, 15, 30, 60, 90, and 120 min after gavage. Serum glucose concentrations were determined using the Accu-Chek^®^ Performa test (Roche, Switzerland) and the area under the glucose–time curve was calculated.

### 2.4. Tissues and Samples

Mice were anesthetized and blood samples collected in EDTA-coated tubes. The serum was used for liver alanine aminotransferase (ALT), aspartate aminotransferase (AST), HDL-cholesterol, and triglyceride (TG) determination. The livers were excised, weighed, and either fixed in buffered 4% paraformaldehyde or frozen for further TG determination and RNA extraction. The proximal ileum and colon were cut into two pieces. One piece was flushed, opened longitudinally, cut into 2 cm sections, and fixed in 4% paraformaldehyde and the other was frozen for later RNA extraction. Fecal samples were collected by proper removal from the colon for gut microbiota analysis. Cecal contents were collected in two separate tubes and frozen. Brown adipose tissue (BAT) and WAT were cut into two pieces: one piece was fixed in 4% paraformaldehyde for histological investigation and the other frozen for later RNA extraction. All samples were stored at −80 °C until use.

### 2.5. Measurement of Liver Triglycerides and Plasma Transaminases

TGs were extracted using an Abcam Triglyceride Assay Kit-Quantification (Cambridge, UK) and measured with a Mithras LB940 instrument (Berthold Technologies, France SAS). The TG level is expressed in nmol per milligram of liver. Transaminases (ALT and AST) were assessed using a spectrophotometric method (Olympus, AU400).

### 2.6. Liver and Adipose Tissue Histology

Liver and adipose tissues (BAT and WAT) were fixed overnight in 4% paraformaldehyde and embedded in paraffin. Liver, WAT, and BAT paraffin sections (3 μm thick) were stained with hematoxylin and eosin (H&E). Images were digitally captured from scanned slides using NanoZoomer 2.0-RS and NDP.view2 software (Hamamatsu, Japan). To determine the adipocyte diameter, 5 fields/slide (×200 magnification) were analyzed for each mouse using NDP.view2 software (Hamamatsu, Japan). For ratio of BAT lipid droplets quantification, we specifically developed an image-processing macro with the Image-J^®^ software (https://imageJ.nih.gov, accessed on 22 October 2021), using set threshold with defined values to limit measure to region of interest area measurement (lipid droplet or HE cytoplasmic staining). 4 fields/slide were analyzed for 3 mice per group.

### 2.7. RNA Extraction and Quantification

Mouse livers, BAT, and WAT were disrupted in Qiazol solution. Ilea were disrupted using an MP Biomedicals FastPrep homogenizer. Total RNA was extracted using a Qiagen RNeasy Lipid Tissue Mini Kit (Courtaboeuf, France). The RNA integrity number (RIN) was determined using an Agilent Bioanalyzer 2100 system with the RNA 6000 Nano Labchip kit. Samples used had a RIN of 8 for liver tissues and a RIN of 7 for gut tissues, BAT, and WAT. For cDNA synthesis, 5 µg of each total RNA sample was reverse transcribed. A 6 µL mix containing 5 µg RNA, random hexamers (Roche Diagnostics, Meylan, France) and 10 mM dNTP mix (Invitrogen, Carslbad, CA) was prepared for each sample. Mixtures were heated at 65 °C for 5 min, cooled on ice, and then an 8 µL reaction mix containing 1 µL M-MuLv RT (Invitrogen), 4 µL 5× Buffer (Invitrogen), 2 µL 0.1 M dithiothreitol (Invitrogen), and 1 µL Protector RNase Inhibitor (40 U/µL; Invitrogen) was added. The reaction conditions were 10 min at 25 °C, 50 min at 50 °C, and 15 min at 70 °C.

### 2.8. Gene Expression Analysis by Quantitative PCR

Real-time qPCR was performed in a Light Cycler 480 (Roche Diagnostics) using the LC FastStart DNA Master SYBR Green I kit (Roche Diagnostics). Amplification was initiated with an enzyme activation step at 95 °C for 10 min, followed by 40 cycles consisting of a 20 s denaturation step at 95 °C, a 15 s annealing step at the temperature appropriate for each primer, and a 10 s elongation step at 72 °C. The primer sequences used to amplify the target cDNA are listed in Appendix A. Data were analyzed using LC 480 Software (Roche Diagnostics). Relative gene expression was normalized to that of the 18S or GAPDH reference gene.

### 2.9. Bacterial DNA Extraction and Analysis of the Gut Microbiota by 16S Ribosomal RNA Sequencing

Bacterial DNA was extracted from feces using a Qiagen QIAamp DNA Stool Mini Kit (Courtaboeuf, France) following disruption with an MP Biomedicals FastPrep homogenizer. The composition of the fecal microbiota was analyzed using Illumina MiSeq technology targeting the 16S ribosomal DNA V3-V4 region in the paired-end mode (2 × 300 base pair) (GenoToul, Toulouse), as described previously [17]. The non-chimeric sequences were then clustered into operational taxonomic units (OTUs) at 97.0% sequence similarity using a closed reference-based picking approach with UCLUST software against the Silva (v132) database [19]. After rarefaction at 18,000 reads per sample, bacterial alpha diversity was estimated using the Shannon Index. OTUs with a prevalence <5% were removed from the analysis. Analyses using R software v2.14.1 were restricted to merged OTUs with the same taxonomic assignment. Results are presented as the mean ± SEM. The Wilcoxon test was used to assess statistical significance of the bacterial composition between the samples. Associations were considered to be significant after a false-discovery rate (FDR) correction of the *p*-value (*q* < 0.05). Beta diversity was assessed using weighted and unweighted UniFrac distances. The weighted UniFrac metric is weighted by the difference in the abundance of OTUs from each community, whereas unweighted UniFrac only considers the absence/presence of the OTUs, providing different information. Both are phylogenetic beta diversity metrics. Potential links between the various groups of mice and bacterial microbial profiles were investigated by performing an ANOSIM test with 10,000 permutations on the beta diversity metrics described above. Linear discriminative analysis (LDA) effect size (LEfSe) analysis was performed to identify the taxa displaying the largest differences in abundance in the microbiota between groups [20]. Only taxa with an LDA score >2 and a significance of *p* < 0.05, as determined using Wilcoxon signed-rank tests, are shown.

### 2.10. Chemicals, Reagents, and Sample Preparation for GC/MS Analysis

Reference and Stable isotope-labeled compounds (acetate, propionate, butyrate, isobutyrate, valerate and isovalerate, acetate-D3, propionate-D2, butyrate-13C2, and valerate-D9) were purchased from Sigma-Aldrich (Saint Quentin Fallavier, France). Analytical grade NaOH, propan-1-ol, pyridine, hexane, and propylchloroformate (PCF) were also purchased from Sigma-Aldrich. Deionized water was prepared using a Milli-Q Elix system fitted with a LC-PaK and a 0.22 μm MilliPak filter (Merck Millipore, Guyancourt, France). The protocol was modified based on the method of Zheng et al. [21]. Extraction steps were carried out at 4 °C to avoid the loss of SCFA species. Mouse fecal samples (30 mg) were suspended in 1.5 mL of a 0.005 M NaOH solution containing internal standard mix and ceramic beads. Samples were homogenized at 6500 rpm for 3 × 20 s using a Prescellys^®^ Evolution device (Bertin Technologies, Montigny-le-Bretonneux, France). Supernatants (300 µL) were collected and transferred to 5 mL glass tubes. Then, 500 µL of a propanol/pyridine mix (3:2 *v*/*v*) was added and the tubes vortexed, followed by two successive additions of 50 µL PCF and vortexing. The mixture was sonicated and centrifuged at 2000× *g* at 4 °C for 5 min and 200 µL of the organic phase transferred to GC/MS vials before their injection. SCFAs in fecal samples were quantified by gas chromatographic/mass spectrometry using an ISQ LT™ equipped with a Triplus RSH (Thermo Fisher Scientific, Illkirch, France). A fused-silica capillary column with a (5%-phenyl)-methylpolysiloxane phase (DB-5 ms, J&W Scientific, Agilent Technologies Inc., Santa Clara, CA, USA) of 50 m × 0.25 mm i.d. coated with a 0.25 µm-thick film was used. The temperatures of the front inlet, MS transfer line, and electron impact ion source were set to 260 °C, 290 °C, and 230 °C, respectively. Helium was supplied with carrier gas at a flow rate of 1 mL/min. The oven temperature was initially set to 50 °C for 1.5 min. The temperature was raised to 70 °C at 8 °C/min and then to 85 °C at 6 °C/min. The temperature was then successively elevated to 110 °C at 22 °C/min and 120 °C at 12 °C/min. The oven temperature was finally elevated to 300 °C at 125 °C/min and the temperature held for 3 min. The run time was 15 min in targeted SIM mode using the mass list shown in Appendix A. The injected sample volume was set to 1 µL in split mode with a 20:1 ratio. Data processing was performed using the manufacturer’s Xcalibur^®^ software (version 3.0, Thermofisher Scientific, Illkirch, France).

### 2.11. Statistical Analyses

Results are presented as the mean ± SEM. The non-parametric Kruskal–Wallis test with Dunn’s multiple comparison post hoc test or the Mann–Whitney test was used to compare the means of groups as appropriate (Graphpad Prism 8.0a, Graphpad Software Inc, La Jolla, CA, USA). A *p* value < 0.05 was considered statistically significant. * *p* < 0.05, ** *p* < 0.01, *** *p* < 0.001.

## 3. Results

### 3.1. Pectin Prevents Liver Steatosis in HFD-Fed Mice and Induces Morphological Changes in White Adipose Tissue and Modifications of the Intestinal Microbiota

Mice were fed an HFD or HFD^2^ for 16 weeks. Pectin had no effect on dietary intake or weight gain (Appendix A). Mice fed an ND^2^ showed no differences in the liver relative to mice fed an ND (Figure 1 and Appendix A). Pectin prevented the development of steatosis in HFD-fed mice, as shown by histological analysis and TG quantification (Figure 1a), leading to a decrease in the liver/body weight ratio (Appendix A), without improvement in ALT (Figure 1b). In addition, we found that mice fed an HFD^2^ showed a decrease in inflammatory gene expression (Figure 1b,c), but this did not reach statistical significance. At the daily dose of pectin used, there was neither improvement in glucose sensitivity nor plasma post-prandial lipids, regardless of the diet (Appendix A). Consistent with the absence of weight loss in HFD^2^-fed mice, there was no change in the weight of epididymal WAT (Figure 1d and Appendix A). However, histological analysis showed a significant decrease in the semi-quantified adipocyte diameter in HFD^2^-fed mice (Figure 1e). This lower size of adipocytes was associated with an increase in Cidea mRNA levels, without a significant increase in those of UCP1 or CPT1 (Figure 1f), markers of the BAT function, suggesting a browning of WAT [22,23]. Of note, such morphological changes in WAT were also observed in ND^2^-fed mice relative to ND-fed mice (Appendix A). Histological analysis of BAT showed the lowering of lipid droplet size in HFD^2^-fed mice (Figure 1g) which was quantified by calculating the lipid droplet surface compared to the surface of cytoplasm. Nevertheless, there was no significant increase in UCP1, CPT1, or Cidea mRNA levels (Figure 1h).

These metabolic changes induced by pectin were associated with modifications of the IM, as shown by principal coordinate analysis (Figure 2a). Pectin decreased the relative abundance of Firmicutes independently of diet (Figure 2b) and specifically increased that of Bacteroidetes in HFD^2^-fed mice (Figure 2c,d). Pectin also increased the abundance of S24_7, *Prevotellaceae*, and *Turicibacteraceae* and, conversely, decreased the abundance of *Desulfovibrionaceae* and *Ruminococcus* in HFD^2^-fed mice relative to HFD-fed mice. Interestingly, at this dose of pectin, we observed a decrease in the relative abundance of proteobacteria independently of diet (Figure 2b). The effect of pectin on the IM was dose dependent, as a lower amount of pectin resulted in dampened changes in IM composition. This could explain why a pectin dose of 0.4% associated with the HFD was not sufficient to improve liver steatosis and only slightly modified IM composition (Appendix A).

Similarly, a curative treatment with 2% pectin applied after the development of obesity and liver steatosis (Figure 3a) did not improve these factors in obese mice, as shown by the quantification of steatosis and transaminases (Figure 3b,c). This result suggests that two months of pectin treatment is not sufficient to reverse HFD-induced liver steatosis without other dietary modifications. However, WAT browning was improved by two months of curative pectin treatment (Figure 3d–f).

Moreover, curative pectin induced a number of changes in the intestinal bacteria species similar to those observed with the preventive pectin treatment (Figure 4). Indeed, among these modifications, we observed an increase in the abundance of *Bacteroides* and a decrease in that of *Desulfovibrionaceae* and *Ruminococcaceae*. Moreover, pectin also decreased the level of proteobacteria in pectin-treated mice (HFD + HFD^2^) (Figure 4). Although there was no improvement in liver steatosis, pectin increased IM diversity in HFD + HFD^2^-fed mice relative to HFD + HFD-fed mice (Appendix A).

### 3.2. The Gut Microbiota Contributes to the Beneficial Effects of Pectin in Obese Mice

We investigated whether the IM modifications induced by pectin played a major role in the improvement of liver steatosis by collecting the IM of mice fed an HFD supplemented or not with pectin and performing fecal microbiota transfer (FMT). We transplanted the IM from HFD obese mice (FMT^HFD^ + HFD) or HFD obese mice treated with pectin (FMT^HFD2^ + HFD) to control mice as a preventive treatment before the induction of obesity (Figure 5a). Similarly, we transplanted the IM, after the induction of obesity as a curative treatment, identified as HFD+ FMT^HFD^ or HFD + FMT^HFD2^ (Figure 5b). As observed in HFD-fed mice, with or without pectin, there were no changes in dietary intake, weight gain, WAT accumulation, ALT levels, glucose sensitivity, or brown adipocyte morphogenesis for either preventive or curative FMT (Appendix A). There were also no significant changes in terms of steatosis for FMT as a preventive treatment (Figure 5c). Conversely, curative FMT decreased steatosis, as shown by hepatic TG content and histological analysis (Figure 5d). There was no effect on liver inflammation.

Curative and preventive FMT of the IM from HFD-fed mice fed pectin were both able to induce metabolic changes in WAT, as shown by the decrease in adipocyte size and the expression of mRNA genes involved in browning (Figure 5e–h). In addition, both curative and preventive FMT of the IM from mice fed pectin induced UCP1 expression (Figure 5g,h). However, we found discrepancies between the expression of CPT1 and Cidea, which increased in mice receiving preventive FMT and decreased in those receiving curative FMT. Moreover, WAT browning was associated with a slight decrease in WAT inflammation, as shown by the decrease in CCL-2 expression in mice receiving curative FMT from mice treated with pectin (Figure 5g,h). Thus, the FMT from mice treated with pectin was sufficient to induce metabolic changes in obese recipient mice.

Principal coordinate analysis of the IM showed that preventive and curative FMT both efficiently modified the IM of recipient mice receiving the pectin-modified IM (Figure 6a,b). Thus, we observed a significant increase in the abundance of Bacteroidetes and a significant decrease in that of *Epsilonbacteraeota* in both preventive and curative FMT. Aside from these changes in phyla abundance, the IM composition was different depending on whether preventive or curative FMT was used (Figure 6a,b). Preventive FMT with the IM from HFD^2^-fed mice increased the relative abundance of *Muribaculum* and *Ruminococcus* and decreased that of *Peptostreptococcaceae*, *Clostridium stricto sensu*, *Ruminococcus gnavus*, *Romboutsia*, and *Paraclostriudum*. Curative FMT with the IM from HFD^2^-fed mice increased the relative abundance of *Prevotellaceae*, *Alloprevotella*, *Streptococcus*, and *Ruminococcus torques* and decreased that of *Ruminococcaceae*, *Mycoplasmataceae*, and *Helicobacteraceae*, including that of *Helicobacter*.

Specific comparisons of the IM from mice receiving preventive or curative FMT from mice treated with pectin showed a specific increase in the relative abundance of enterobacteria, including *Escherichia Shigella*, and an increase in that of Helicobacter in mice receiving specifically preventive FMT (FMT^HFD2^ + HFD), associated with an absence of liver improvement, in contrast to mice receiving curative FMT (Figure 7a).

Of note, the relative abundance of *Lactobacillus* also increased in these mice. These differences could explain the discrepancies observed in short-chain fatty acid (SCFA) quantification after preventive or curative FMT (Figure 7b). Indeed, analysis of the SCFAs of the cecal content showed that preventive FMT induced a decrease in the concentration of the branched SCFAs, isobutyrate, and isovalerate, whereas curative FMT induced an increase in acetate, butyrate, and propionate concentrations, without significant changes in the total amount of SCFA (Figure 7b). Overall, these modifications of the SCFAs associated with the changes in the IM could explain the more beneficial effect of the curative FMT relative to the preventive intervention.

## 4. Discussion

The IM plays a role in the development of metabolic diseases, such as obesity, being overweight, and insulin resistance, as well as NAFLD [3,4]. Therefore, developing strategies to manage IM composition are of particular interest. Dietary fiber and prebiotics have already shown their beneficial effect in metabolic function [13]. Thus, identifying bacteria and/or bacterial metabolites involved in their beneficial effects is of major interest for new therapeutic avenues. Apple-derived pectin is a soluble fiber, and 7.5% pectin added to the diet has been shown to improve liver steatosis in obese mice [14]. However, high doses of fiber may be associated with poor tolerance, including bloating and abdominal distension [24]. We therefore tested the efficacy of low doses of pectin in NAFLD. Moreover, we used FMT to transfer the IM reshaped by a pectin diet into recipient mice to identify the specific role of the IM.

Pectin was able to prevent liver steatosis in obese mice at a low concentration of 2% (0.06g of pectin/30 g of mouse). Conversely, the addition of a low dose of pectin to an HFD was not sufficient to reverse liver steatosis after two months of treatment of HFD-fed mice. This result suggests that the dose of pectin could be lower for a curative treatment. This is supported by the use of a very low dose of pectin (0.4%), which was not able to improve liver steatosis, used as a preventive treatment in association with the HFD. However, the curative pectin treatment at 2% induced morphological changes in WAT, supporting browning of the tissue. The discrepancies between the beneficial effect on WAT and the absence of a significant effect in the liver may be related to the insufficiently long duration of treatment. Indeed, at least two months of treatment are necessary to alleviate metabolic disorders and steatosis in mice after a shift from a high fat to a normal diet [25].

Soluble fiber consumption can lead to abdominal discomfort or even intestinal pain, rendering its use at a high dose difficult. To overcome such side-effects, the use of intestinal bacteria to mediate the effect of pectin could be of interest. We show that preventive treatment with 2% pectin induced modifications of the IM. Thus, the IM of HFD-fed mice was modified by preventive pectin treatment, which promoted an increase in the relative abundance of certain bacteria that produce SCFAs, including Bacteroides (producer of acetate and propionate) and Turicibacter. The abundance of Turicibacterales, which produce butyrate, has been shown to inversely correlate with the NAFLD phenotype, and has been found to be reduced in NAFLD patients [26]. Interestingly, there was no increase in the abundance of proteobacteria at this dose of pectin, in contrast to previous studies using higher doses [15,27].

We carried out FMT to determine the role of the IM in the pectin effect, either as a preventive or curative treatment, after four months of an HFD diet. Under these conditions, the occurrence of liver damage was not alleviated by preventive FMT but was attenuated by curative FMT. Nevertheless, browning of the WAT was observed with both preventive and curative FMT. The IM is a major modulator of host metabolic homeostasis [28] and it has been shown that modulation of the IM by antibiotics, exposure to cold, or intermittent fasting can improve metabolic disorders, including the browning of WAT in obese mice [29,30,31]. The IM of FMT recipient mice showed an increase in the relative abundance of Bacteroidetes, in accordance with the increase in that of Bacteroides induced in the IM of donor mice. Preventive FMT induced an increase in the relative abundance of *Muribaculum* and *Ruminococcus*, whereas curative FMT induced an increase in that of *Alloprevotella* and *Ruminococcus torques*, both species that produce SCFAs [32,33].

SCFAs are major products derived from the microbial fermentation of dietary fiber and have a broad impact on host physiology [11]. Butyrate, acetate, and propionate participate in the improvement of hepatic steatosis, either by activating the GPR41 and GPR43 receptors located within the target tissues (adipose tissue, intestine, liver) or by directly acting locally without binding to their receptors [2]. We observed an increase in UCP1 mRNA levels, both in preventive and curative FMT. Conversely, we found differential modulation of CPT1 mRNA levels between the two approaches, which may be related to the mitochondrial import of free fatty acids and the modulation of ß-oxidation, as well as those of Cidea, which correlate with liver steatosis [34]. These discrepancies could be explained by differences in the modulation of SCFAs between preventive and curative FMT. Indeed, preventive FMT decreased isobutyrate and isovalerate levels without changing those of butyrate, acetate, and propionate. A similar protective association between isobutyrate and isovalerate has been shown in HFD-fed mice receiving FMT from obese mice without metabolic syndrome [3]. However, these branched fatty acids have been associated with insulin resistance and the development of metabolic diseases [35]. The relative abundance of *Muribaculum* and *Ruminococcus* increased in mice receiving preventive FMT. These bacteria are associated with the production of propionate, acetate, and butyrate, although we did not find any increase in the concentration of these SCFAs in the cecum. Conversely, curative FMT induced increased levels of butyrate, propionate, and acetate without modification of those of isobutyrate or isovalerate. Treatment with guar gum, a soluble fiber that shares similar properties with pectin, or SCFA (acetate, butyrate, or propionate) administered as a preventive treatment during an HFD also results in decreased steatosis and WAT remodeling, similarly to our results [36,37]. However, it has been shown that the beneficial action of SCFAs resulting from supplementation of the diet with fiber does not correlate with SCFA concentrations in the cecum but with their levels in the circulation [36]. It would be therefore useful to measure the concentration of these SCFAs in the plasma, in addition to the cecal assay.

Our data are in accordance with the literature on the beneficial role of pectin in NAFLD models. The IM modified by pectin appears to have a significant effect on HFD-induced injuries, although not all parameters were restored relative to direct treatment with pectin. Thus, FMT induced changes in the IM that allowed more efficient remodeling of the bacterial ecosystem than direct consumption of the fiber itself when the diet was not altered. This suggests that dietary intervention could be improved by the concomitant use of probiotics, which may allow a decrease in the amount of fiber consumed and the associated intestinal discomfort. However, combinations of prebiotics, including pectin or inulin, could also overcome the observed digestive side-effects and increase bacterial diversity and the production of beneficial metabolites [38,39]. In vitro studies have shown that a mixture of five types of fiber, including inulin and pectin, and a panel of Lactobacillus could help in the design of such a prebiotic treatment [39,40]. Overall, these observations are supported by a recent publication showing that fermented food can increase IM diversity and decrease inflammation more effectively than a high-fiber food diet [41].

Thus, functional foods containing dietary fiber offer hope as an alternative treatment for NAFLD. Nevertheless, dietary interventions in humans need to be personalized to find a treatment with the least digestive side-effects and the most efficient modification of IM function for each individual. As the changes in the IM after prebiotic and probiotic interventions depend on its initial composition [16,42], personalized nutritional interventions may become one of the major challenges in the follow-up of patients in the future.

## Figures and Tables

**Figure 1 nutrients-13-03725-f001:**
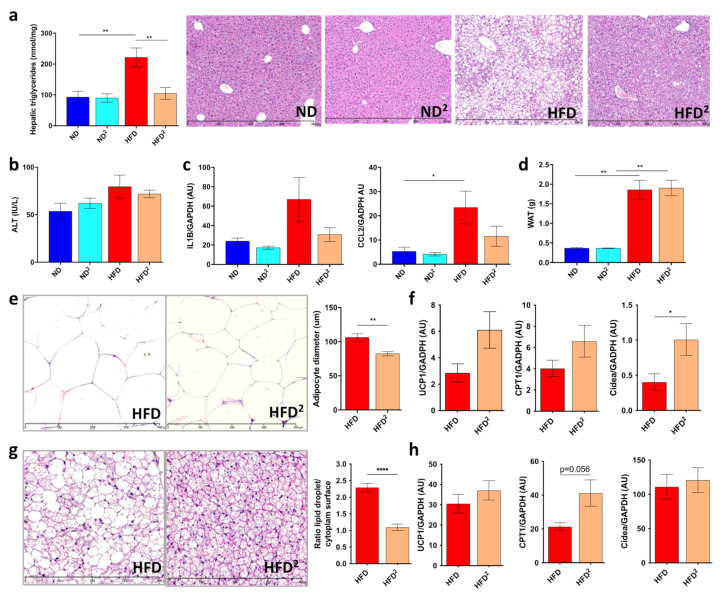
Pectin improves liver steatosis in mice fed a high-fat diet. Mice were fed a normal diet (ND), an ND with 2% pectin (ND^2^), a high-fat diet (HFD), or an HFD with 2% pectin (HFD^2^) for 16 weeks: (**a**) hepatic triglyceride content and histological examination of liver tissue sections stained with hematoxylin-eosin (scale 800 μm). (**b**) Plasma ALT. (**c**) Quantification of CCL2 and IL1β mRNA in the liver by RT-qPCR. (**d**) Weight of epididymal white adipose tissue (WAT). (**e**) WAT section stained with hematoxylin-eosin (scale 400 μm) and histomorphometric analysis of adipocyte diameter (μm). (**f**) Quantification of UCP1, CPT1, and Cidea mRNA in WAT by RT-qPCR. (**g**) Brown adipose tissue (BAT) section stained with hematoxylin-eosin (scale 400 μm) and ratio of lipid droplets surface and cytoplasm surface. (**h**) Quantification of UCP, CPT1, and Cidea mRNA by RT-qPCR in BAT. Data represent the mean ± SEM of eight mice. * *p* < 0.05, ** *p* < 0.01, **** *p* < 0.0001.

**Figure 2 nutrients-13-03725-f002:**
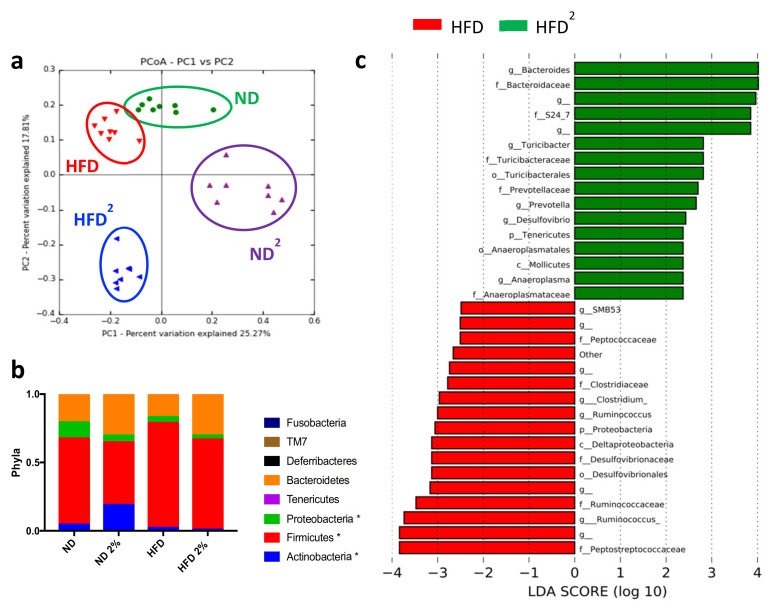
Modifications of the intestinal microbiota by a high-fat diet with 2% pectin. Mice were fed a normal diet without or with 2% pectin (ND, ND^2^) or a high-fat diet without or with 2% pectin (HFD, HFD^2^) for 16 weeks. Microbiota analysis was then performed by 16S sequencing: (**a**) unweighted Unifrac distances showing a difference in the composition of fecal microbiota between groups (*p* < 0.001, R = 0.58, ANOSIM test, 10,000 permutations, using the first 5 PC): green = ND, purple = ND^2^; red = HFD, and blue = HFD^2^. (**b**) Histogram showing the relative abundance between phyla (* *p* < 0.05, represent statistical difference after Bonferonni post-hoc test). (**c**) LDA effect size (LEfSe) for the taxa enriched in HFD (red) and HFD^2^ mice (green); only taxa with a, LDA score >2 and *p* < 0.05, determined by the Wilcoxon signed rank test, are shown. There were eight mice per group.

**Figure 3 nutrients-13-03725-f003:**
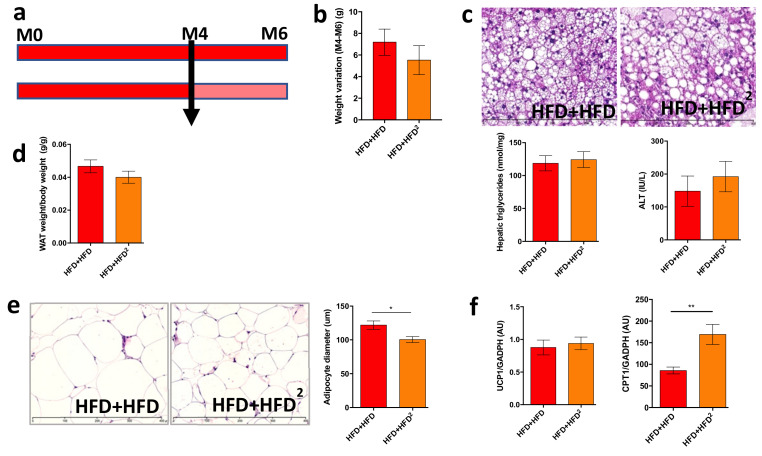
Pectin supplementation in HFD mice induces WAT browning but is not sufficient to improve liver steatosis. Mice were fed an HFD for 16 weeks and received pectin supplementation in the HFD (2%) from week 16 to week 24 as a curative treatment (HFD + HFD^2^) versus mice who did not receive pectin (HFD + HFD): (**a**) design of the mouse experiment. (**b**) Change in body weight between 16 (M4) and 24 weeks (M6). (**c**) Histological images of a liver tissue section stained with hematoxylin-eosin (scale 400 μm), hepatic triglyceride content, and plasma ALT. (**d**) White adipose tissue (WAT) weight/body weight ratio. (**e**) WAT section stained with hematoxylin-eosin (scale 400 μm) and histomorphometric analysis of adipocyte diameter (μm). (**f**) Quantification of mRNA expression by RT-qPCR of UCP1 and CPT1 in WAT. Data represent the mean ± SEM of eight mice. * *p* < 0.05, ** *p* < 0.01.

**Figure 4 nutrients-13-03725-f004:**
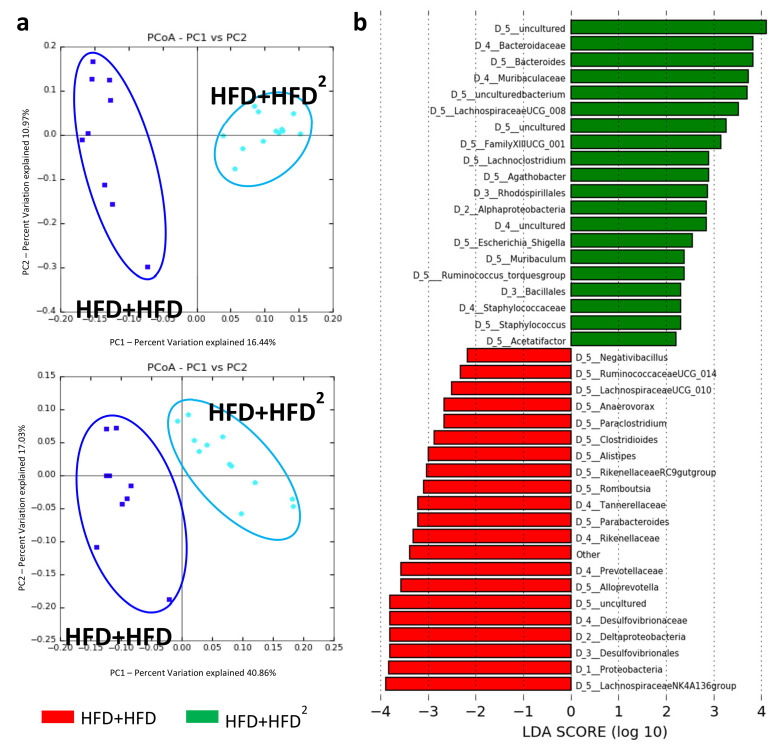
Modification of the IM of HFD mice following pectin supplementation. Mice were fed an HFD for 16 weeks and received pectin supplementation (2%) from week 16 to week 24 as curative treatment (HFD+HFD^2^) versus mice not receiving pectin (HFD + HFD): (**a**) unweighted (left, *p* = 0.02) and weighted (right, *p* = 0.01) Unifrac distances showing differences in IM composition and relative OTU abundance between HFD + HFD (dark blue) and HFD + HFD^2^ (light blue) mice. (**b**) LDA Effect Size (LEfSe) for the taxa enriched in HFD + HFD (red) or HFD + HFD^2^ (green) mice. Only taxa with a LDA score >2 and *p* < 0.05, determined by the Wilcoxon signed rank test, are shown. Data represent analysis of the IM of 9 or 12 mice.

**Figure 5 nutrients-13-03725-f005:**
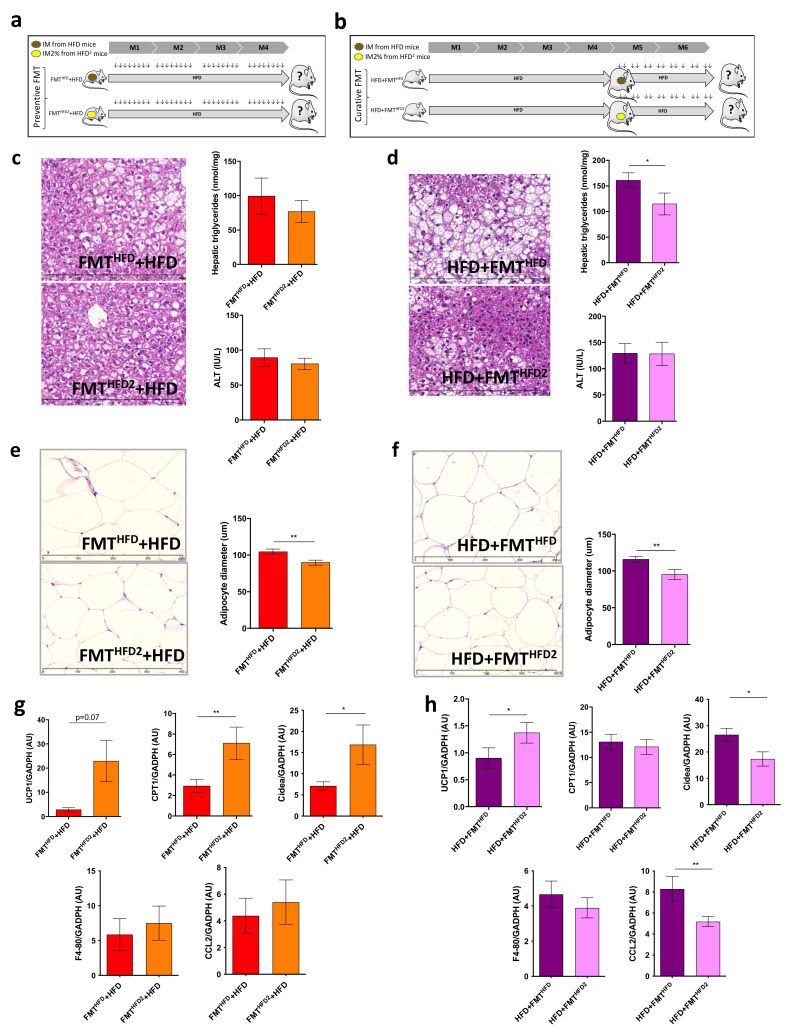
FMT from mice fed an HFD with pectin is sufficient to induce WAT browning in recipient HFD-fed mice: (**a**) experimental setup showing the preventive fecal microbiota transplantation (FMT). Mice received FMT from donor mice fed an HFD (FMT^HFD^ + HFD) or donor mice fed an HFD^2^ (FMT^HFD2^+ HFD). Preventive FMTs were performed twice a week during the 16 weeks of the HFD. (**b**) Experimental setup showing the curative FMT performed after 16 weeks of an HFD and twice a week during eight supplementary weeks of an HFD. Obese mice received a curative FMT before eight supplementary weeks of an HFD from donor mice fed an HFD (HFD + FMT^HFD^) or donor mice fed an HFD^2^ (HFD + FMT^HFD2^). (**c**,**d**) Histological images of a liver tissue section stained with hematoxylin-eosin (scale 400 μm) and hepatic triglyceride content. (**e**,**f**) White adipose tissue (WAT) section stained with hematoxylin-eosin (scale 400 μm) and histomorphometric analysis of adipocyte diameter (μm). (**g**,**h**) Quantification of mRNA expression in WAT by RT-qPCR. Data represent the mean ± SEM of 8 or 12 mice. * *p* < 0.05, ** *p* < 0.01.

**Figure 6 nutrients-13-03725-f006:**
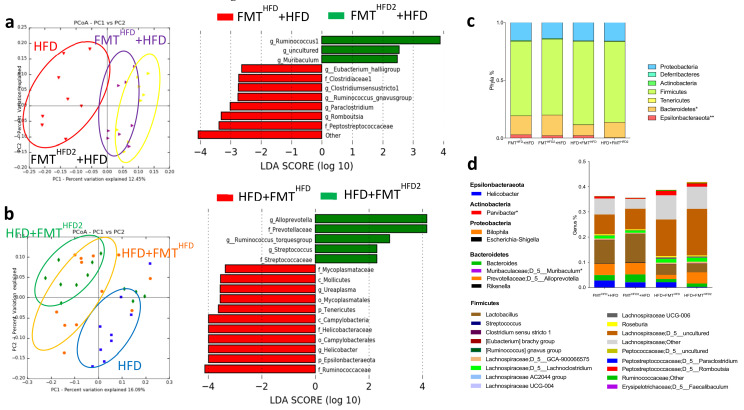
Curative and preventive FMT induce modifications of IM composition. Mice received a preventive FMT before 16 weeks of HFD from donor mice fed an HFD (FMT^HFD^ + HFD) or donor mice fed an HFD2 (FMT^HFD2^+ HFD). Obese mice received a curative FMT before eight supplementary weeks of HFD from donor mice fed an HFD (HFD + FMT^HFD^) or donor mice fed an HFD^2^ (HFD + FMT^HFD2^): (**a**) unweighted Unifrac distances showing differences in IM composition between the groups: red = HFD (4 months of diet), yellow = FMT^HFD2^ + HFD, purple = FMT^HFD^ + HFD (*p* = 0.01) (left panel), and LDA effect size (LEfSe) for the taxa enriched in FMT^HFD^ + HFD (red) or FMT^HFD2^ + HFD (green) mice (right panel). Only taxa with a LDA score >2 and *p* < 0.05, determined by the Wilcoxon signed rank test, are shown. (**b**) Unweighted Unifrac distances showing differences in IM composition between the groups: blue = HFD (6 months of diet), green = HFD + FMT^HFD2^, orange = HFD + FMT^HFD^; (*p* = 0.08) (left panel) and LDA Effect Size (LEfSe) for the taxa enriched in HFD + FMT^HFD^ (red) or HFD + FMT^HFD2^ (green) mice (right panel). Only taxa with a LDA score >2 and *p* < 0.05, determined by the Wilcoxon signed rank test, are shown. (**c**) Relative abundance of bacteria at the phyla and (**d**) genus level. Data represent the mean ± SEM of 9 or 12 mice. * *p* < 0.05, ** *p* < 0.01.

**Figure 7 nutrients-13-03725-f007:**
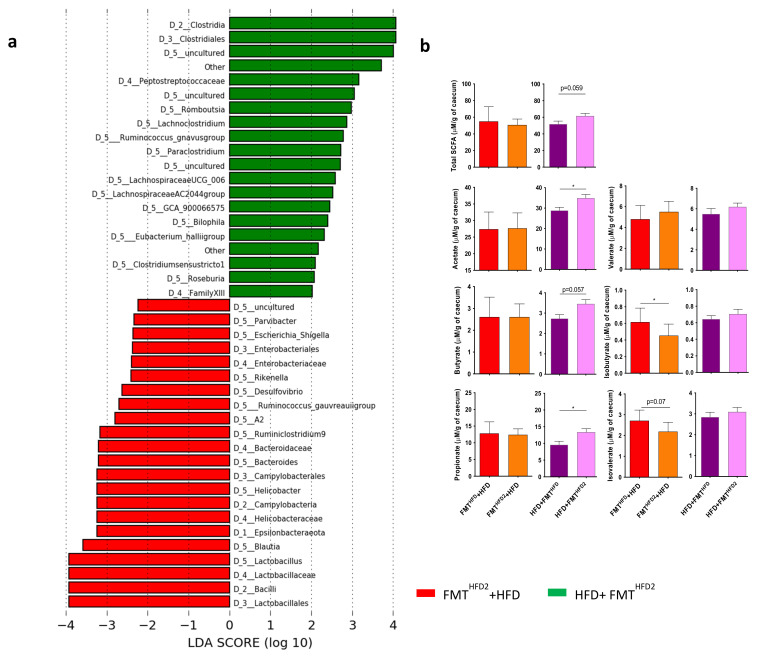
Curative and preventive FMT induce modifications of the IM and the proportion of SCFAs in cecal content. Mice received a preventive FMT before 16 weeks of HFD from donor mice fed an HFD (FMT^HFD^ + HFD) or donor mice fed an HFD^2^ (FMT^HFD2^ + HFD). Obese mice received a curative FMT before eight supplementary weeks of HFD from donor mice fed an HFD (HFD + FMT^HFD^) or donor mice fed an HFD^2^ (HFD + FMT^HFD2^): (**a**) LDA effect size (LEfSe) for the taxa enriched in mice receiving the IM of pectin-treated mice by preventive FMT^HFD2^ + HFD (red) or curative HFD + FMT^HFD2^ (green) treatment. Only taxa with a LDA score > 2 and *p* < 0.05, determined by the Wilcoxon signed rank test, are shown. (**b**) Cecal SCFA content induced by preventive and curative FMT (μM/mg of cecal content). Data represent the mean ± SEM of 9 or 12 mice. * *p* < 0.05.

## Data Availability

Data supporting the findings of this study are available from the corresponding author.

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
