# Peer review of "Gut Microbiota Reshaped by Pectin Treatment Improves Liver Steatosis in Obese Mice"

_nutrients, 2021, doi:10.3390/nu13113725_

Round 1

Reviewer 1 Report

Overall, the manuscript is well organized and written. It’s an interesting study, however, some questions need to be considered.

Comments

  1. To explore the roles of changed microbiota by pectin in the improvement of liver injury, the authors transplant fecal microbiota derived from the mice fed by HFD diet with or without pectin supplements. The pectin in the collected fecal pellets may impact the interpretation of the FMT experiments, i.e. the remaining pectin could affect microbiota composition in the recipient mice. If possible, the metabolism of pectin should be added.   
  2. In the methods, the authors also collected fecal materials from ND and ND2 and seem performed FMT also. But there is no result shown in the manuscript. Please revise the text and explain why only used HFD groups for transplantation in the discussion.

Reviewer 2 Report

The manuscript describes the effect of pectin administration on the phenotypes of high fat diet (HFD)-induced NAFLD model mice, with additional experiment of fecal microbiota transfer from mice treated with pectin. The results would be of great interest to readers of this journal. However, there were multiple problems to be solved as listed below:

Specific issue:

  1. There was no evidence of prevention of “liver injury” by 2% pectin in this paper. The pectin treatment decreased hepatic steatosis but not serum transaminase. The term “liver injury” represents liver disfunction with hepatocellular injury, indicated by serum transaminases or some cell death markers (apoptosis, necrosis). Besides, the HFD model in this paper did not induce apparent liver injury as the increase of serum ALT was modest (<2 fold from ND group).
  2. L116-117 – What was counted (or measured) from WAT sections by fluorescent microscope? No fluorescent image was provided. Additionally, the method of adipocyte diameter analysis should be explained in the method section. How many adipocytes and how many fields were analyzed? What kind of software or tool was used?
  3. Figure S1C – Absolute liver weight data should be added. As body weight increased by HFD feeding, relative liver weight (Figure S1C) in HFD and HFD2 groups was biased. The difference in Figure S1C would be of less biological significance.
  4. L217 – The definition of the browning of WAT should be explained with some reference(s) cited. Otherwise, readers cannot understand what is “typical” browning.
  5. L222 – As well as the decreased lipid droplet size, cell size of BAT also looked decreasing in Figure 1g. The authors should measure cell size and add the morphometric data.

Minor points:

Figure S1B ­– Duration of body weight gain measurement? 16 weeks? In addition, body weight at necropsy should also be shown.

Figure 3a – The experimental design in Figure 3a should be explained in the method section, not only in the figure legends. M0, M4, and M6 should be changed to Week 0, 16, and 24, respectively. Same problem on Figure 5a.
